# Time series analysis of daily reported number of new positive cases of COVID-19 in Japan from January 2020 to February 2023

**Ayako Sumi**⬡*

Department of Liberal Arts and Sciences, Division of Physics, Center for Medical Education, Sapporo Medical University, Sapporo, Hokkaido, Japan

* sumi@sapmed.ac.jp

**Data Availability Statement:** The dataset of reported COVID-19 cases analyzed during the current study are contained in Supporting information files (S1 Dataset). The data are also available from ref. [12].

## Abstract

This study investigated temporal variations of the COVID-19 pandemic in Japan using a time series analysis incorporating maximum entropy method (MEM) spectral analysis, which produces power spectral densities (PSDs). This method was applied to daily data of COVID-19 cases in Japan from January 2020 to February 2023. The analyses confirmed that the PSDs for data in both the pre- and post-Tokyo Olympics periods show exponential characteristics, which are universally observed in PSDs for time series generated from non-linear dynamical systems, including the so-called susceptible/exposed/infectious/recovered (SEIR) model, well-established as a mathematical model of temporal variations of infectious disease outbreaks. The magnitude of the gradient of exponential PSD for the pre-Olympics period was smaller than that of the post-Olympics period, because of the relatively high complex variations of the data in the pre-Olympics period caused by a deterministic, nonlinear dynamical system and/or undeterministic noise. A 3-dimensional spectral array obtained by segment time series analysis indicates that temporal changes in the periodic structures of the COVID-19 data are already observable before the commencement of the Tokyo Olympics and immediately after the introduction of mass and workplace vaccination programs. Additionally, the possibility of applying theoretical studies for measles control programs to COVID-19 is discussed.

## Introduction

Since December 2019, a novel coronavirus designated as Severe Acute Respiratory Syndrome Coronavirus 2 (SARS-CoV-2) has rapidly spread around the world, affecting millions of people worldwide; its impact continues today. Waves of cases of this novel coronavirus disease, also known as COVID-19, still occur recurrently, although these waves may well be prevented, and possibly eradicated, in the future. Considerable effort to prevent and eradicate COVID-19 has been expended through COVID-19 surveillance, vaccinations, and theoretical and experimental research [1–4]. Among these efforts, attempts to elucidate the mechanism of the COVID-19 pandemic have been of great interest. Recently, researchers have tried to interpret

**Funding:** AS Grant Number JP22K10529 JSPS KAKENHI https://www.jsps.go.jp/english/e-grants/ NO The funder (JSPS KAKENHI) had no role in study design, data collection and analysis, decision to publish, or preparation of the manuscript.

**Competing interests:** The author has declared that no competing interests exist.

the behavior of the pandemic in terms of deterministic chaos [5–7]. Sapkota et al., reported that the hosting of the Tokyo Olympic Games in Japan between 23 July and 8 August 2021 affected the mechanism of the COVID-19 pandemic in the country [8]. Therefore, the temporal variations of the data before and after the Olympic Games may differ, and examining this point is significant from the standpoint of predicting the COVID-19 pandemic. However, typical approaches cannot fully elucidate the temporal variations of the patterns of the pandemic. This is because the data lengths of reported cases of COVID-19 are very short: in Japan, the data are collected daily, and the number of data points was slightly above 1000 points by February 2023. There is a need for a superior and powerful time series analysis method that can elucidate the temporal variations of a time series even with short data lengths. In previous publications [9–11], we proposed a method that enables us to analyze the time series of COVID-19 cases. In the present study, this method was applied to examining the temporal variations of daily time series data of reported COVID-19 cases for the entire country of Japan. Quantitative elucidation of the COVID-19 pandemic is an important issue in epidemiology today and essential for the development of disease surveillance.

## Methods

### Data

The present study analyzed daily reported number of new positive cases of COVID-19 for the entire country of Japan from 16 January 2020 to 21 February 2023 (1,133 data points). During this period, a total 33,119,744 new positive cases of COVID-19 were reported in Japan, which represents 26.5% of the country's total population of 125 million. The data used in the present study were obtained from the Japan Ministry of Health, Labour, and Welfare COVID-19 Data [12]. The data are indicated in the S1 Dataset.

### Time series analysis

We used a time series analysis consisting of maximum entropy method (MEM) spectral analysis in the frequency domain and least squares method (LSM) in the time domain [9–11,13,14]. The MEM is considered to have a high degree of resolution of spectral estimates compared with other analysis methods of infectious disease surveillance data such as the fast Fourier transform algorithm and autoregressive methods, which require time series of long data lengths [15,16]; therefore, an MEM spectral analysis allows us to precisely determine short data sequences, such as the infectious disease surveillance data used in this study [9–11,13–15]. In the present analysis, we take the reported number of new positive cases of COVID-19 $x(t)$ ($t$: time) to be discrete at $t = k\Delta t$ ($k = 1, 2, 3,\ldots N$), where $\Delta t$ is the equal sampling interval and $N$ is the length of the time series ($\Delta t = 1$ day and $N = 1133$, in the present study).

**MEM spectral analysis.** We assumed that the time series data $x(t)$ (where $t$ is time) were composed of systematic and fluctuating parts [17]:

$$x(t) = \text{systematic part} + \text{fluctuating part.} \tag{1}$$

To investigate temporal patterns of x(t) in the daily time series data, we performed MEM spectral analysis [9–11,13,14]. MEM spectral analysis produces a power spectral density (PSD), from which we obtain the power representing the amount of amplitude of $x(t)$ at each frequency (note the reciprocal relationship between the scales of frequency and period). The MEM-PSD, $P(f)$ (where $f$ represents frequency), for the time series with $\Delta t$, can be expressed

by

$$P(f) = \frac{P_m \Delta t}{\left| 1 + \sum_{k=-m}^{m} \gamma_{m,k} \exp[-i2\pi f k \Delta t] \right|^2},$$ (2)

where the value of $P_m$ is the output power of a prediction-error filter of order $m$ and $y_{m,k}$ is the corresponding filter order. The value of the MEM-estimated period of the $n$th peak component $T_n$ (= $1/f_n$; where $f_n$ is the frequency of the $n$th peak component) can be determined by the positions of the peaks in the MEM-PSD. The MEM-PSD is calculated up to the Nyquist frequency (= $1/(2\Delta t)$), which corresponds to $f$ = 182.5 (1/year) in the present study.

**LSM.** The validity of the MEM spectral analysis results was confirmed by calculating the least squares fitting (LSF) curve pertaining to the original time series data $x(t)$ with MEM-estimated periods. The formula used to generate the LSF curve for $X(t)$ was as follows:

$$X(t) = A_0 + \sum_{n=1}^{N} A_n \cos\{2\pi f_n(t + \theta_n)\}.$$ (3)

The above formula is calculated using the LSM for $x(t)$ with unknown parameters $f_n$, $A_0$ and $A_n$ ($n$ = 1, 2, 3, . . ., $N$), where $f_n$ (= $1/T_n$; $T_n$ is the period) is the frequency of the $n$-th component; $A_0$ is a constant that indicates the average value of the time series data; $A_n$ is the amplitude of the $n$-th component; $\theta_n$ is the phase of the $n$-th component; and $N$ is the total number of components. The reproducibility level of $x(t)$ from the optimum LSF curve was evaluated using Pearson's correlation ($\rho$) with SPSS v. 17.0J software (SPSS, Japan). The expression of Eq (3) is derived from the Fourier transformation of the MEM-PSD as described in the "Discussion" section.

**Preparing the data for analysis.** In Fig 1A, daily reported number of new positive cases of COVID-19 from 16 January 2020 to 21 February 2023 in Japan, $x(t)$, are plotted. Fig 1A' shows the frequency histogram for $x(t)$ (Fig 1A). This histogram deviates from the normal distribution required for conventional spectral analysis. We then introduced the logarithmic transformation of $x(t)$ (Fig 1A). Because the data from 16 January to 10 February 2020 include 17 zero values and are not available for the logarithm transformation, the 15 cases that were reported during this period are ignored and the starting point of the data was re-set to 11 February 2020. As a result, the present study uses the data from 11 February 2020 to 21 February 2023. Logarithm-transformed data for this period are shown in Fig 1B. In the figure, a long-term increasing trend can be observed, with an increase in the spike of reported cases observed at $x(t)$.

To remove the long-term trend of the log-data shown in Fig 1B, PSD, $P(f)$ ($f$: frequency), for the log-data was calculated, and the PSD ($f \leq 1.2$) is displayed in Fig 1C (unit of $f$: 1/year). Therein, the longest period can be observed as the prominent peak at the position of the 5.3-year period. Using this 5.3-year period, we modeled the long-term trend in the COVID-19 pandemic by calculating the LSF curve (Eq (3)) for the entire log-data (Fig 1B). The LSF curve obtained (Fig 1B) expresses the long-term trend of the log-data well.

After the LSF curve was removed from the log-data (Fig 1B), the residual time series data were obtained, as shown in Fig 1D. The frequency histogram of this residual data is shown in Fig 1D', which approximates the normal distribution required for conventional spectral analysis, although a slight difference was observed.

**Segment time series analysis.** We further investigated periodic structures of the residual data (Fig 1D) with segment time series analysis. The residual data (Fig 1D) were divided into several segments and the MEM-PSD for each segment was computed. In this study, each segment represents a 365-day time interval, with the starting point of two consecutive segments

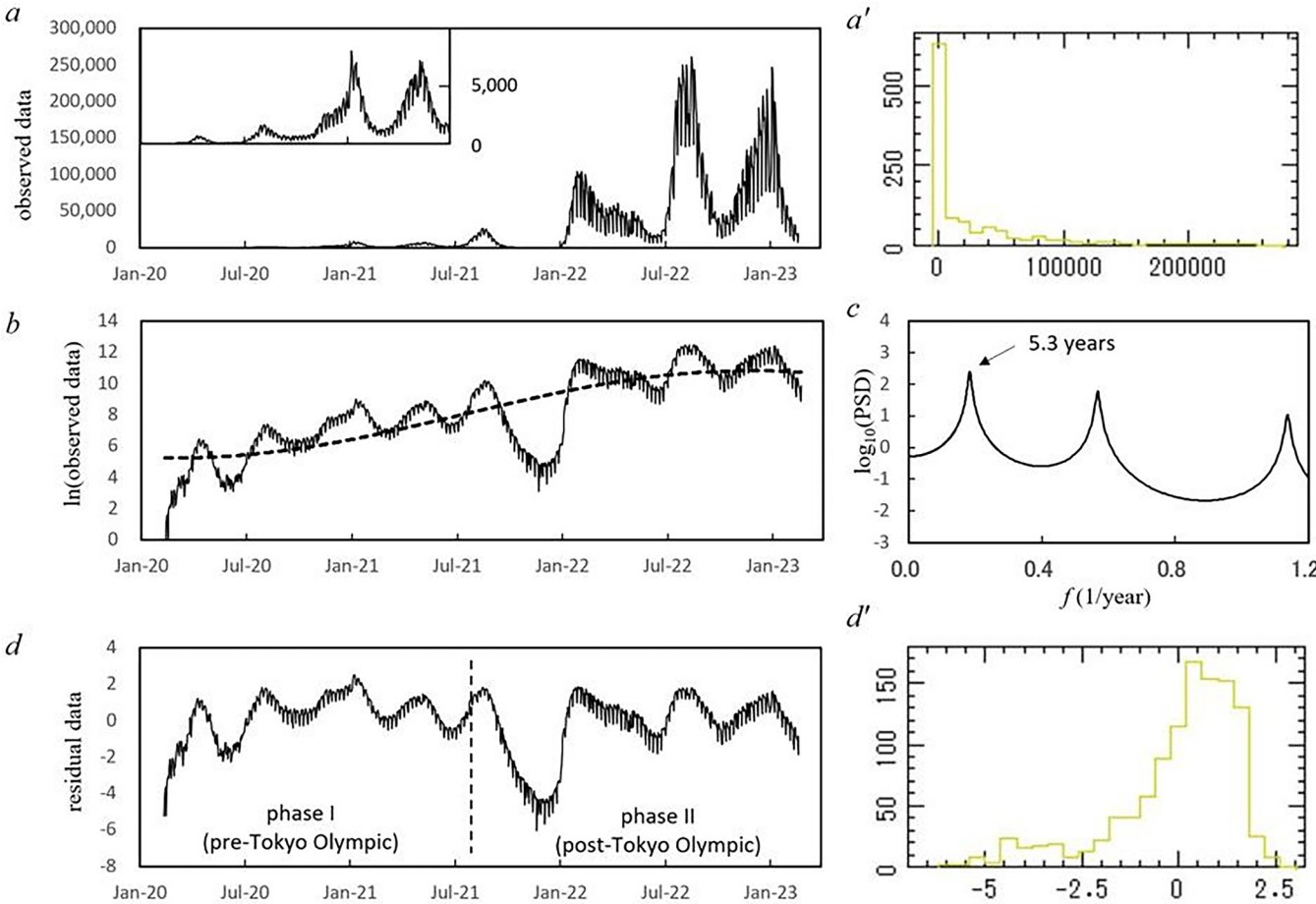

**Fig 1. Daily reported number of COVID-19 new positive cases of Japan from 16 January 2020 to 21 February 2023.** (a) Original data, [inset] enlargement of the original data from 16 January 2020 to 30 June 2021. (a') Histogram of the original data. (b) Logarithm-transformed data of the original data (solid line) and its optimum LSF curve (dashed line). (c) MEM-PSD of the residual data in the low-frequency range ($f \leq 1.2$). (d) Residual data obtained by subtracting the LSF curve from the log-data. (d') Histogram of the residual data. Dashed vertical line in d indicates the boundary of phase I (pre-Tokyo Olympic Games, 11 February 2020–22 July 2021) and phase II (post-Tokyo Olympic Games, 23 July 2021–21 February 2023).

delayed by 6 days. The MEM-PSDs thus obtained were arranged in the order of the time sequence to construct a 3-dimensional (3D) spectral array.

 **Outline of the analysis procedure.** First, MEM spectral analysis was performed for the log-data (Fig 1B) and the long-term period was determined from the PSD for the log-data (Fig 1C). Then long-term trends in the log-data were calculated using the LSF method (Eq (3)) with the MEM-estimated period. This LSF curve (Fig 1B) that corresponded to the long-term trend was removed by subtracting the LSF curve from the log-data, and thus the residual data (Fig 1D) were obtained. Then the MEM-PSDs of phases I and II of the residual data were calculated (Fig 2A and 2B, respectively). Finally, segment time series analysis was performed and the MEM-PSD was obtained for each segment of the residual data (Figs 5 and 6).

## Results

### Temporal variations in the reported number of new positive cases of COVID-19

A closer view of the original data (Fig 1A) from January 2020 to June 2021 is illustrated in the inset of Fig 1A. The figure shows four large waves observed at intervals of about four to five

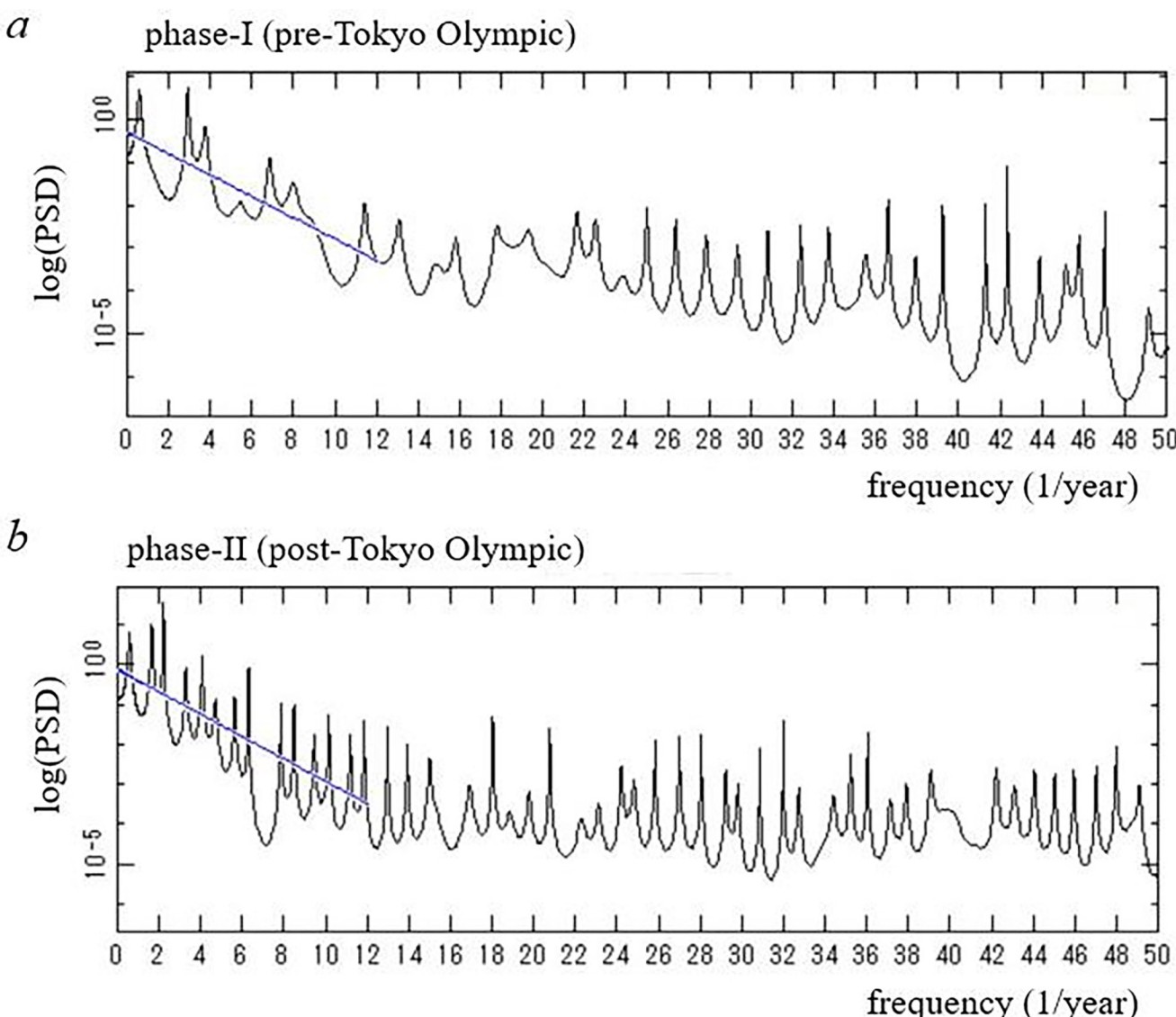

**Fig 2. MEM-PSDs for two ranges of the residual data ($f \leq 50.0$).** (a) Phase I (pre-Tokyo Olympic Games, 11 February 2020–22 July 2021). (b) Phase II (post-Tokyo Olympic Games, 23 July 2021–21 February 2023). In each figure, the blue line in the low-frequency range ($f \leq 12.0$) indicates the regression line with gradient $\lambda$ of Eq (4). See the text for details.

months, peaking in April and July 2020, and January and May 2021. Subsequently, Fig 1A shows four large waves at longer intervals than before, approximately five to six months, with peaks in August 2021, February and August 2022, and January 2023.

## Power spectral density of the time series data

To investigate the effect of hosting the Tokyo Olympic Games on the temporal variations of the COVID-19 pandemic in Japan, the residual data (Fig 1E) were divided into two ranges (phases I and II) in accordance with the starting and ending points of the Tokyo Olympic Games (23 July 2021 and 8 August 2021, respectively): pre-Olympic Games (11 February 2020–22 July 2021) for phase I and post-Olympic Games (23 July 2021–21 February 2023) for

phase II. MEM-PSDs for the residual data of phases I and II were calculated. The semi-log plots of the PSD ($f \leq 50.0$) are shown in Fig 2A and 2B for phases I and II, respectively.

**Gradient of power spectral density.** PSDs for phases I and II are shown in in Fig 2A and 2B, respectively. Therein, the overall trend of each PSD indicates the exponential form

$$P(f) \sim \exp(-\lambda f) \tag{4}$$

until the PSDs level off at the lowest limit determined by the accuracy of the present data, i.e., the number of significant digits in the data. To obtain the magnitude of $\lambda$, the mean power of the PSD was calculated by integrating the PSD over small frequency interval $\Delta f$, that is, the mean power of the PSD is the power in the interval of frequencies $[f, f + \Delta f]$. The line of the PSD gradient is calculated as a regression line against the mean powers in the low frequency range ($f \leq 12.0$), and is drawn in Fig 2A and 2B. The precise value of $\lambda$ was determined using this procedure. The values of $\lambda$ for phases I and II are 0.25 and 0.28, respectively.

**Dominant spectral lines.** In the PSDs for phases I and II (Fig 2A and 2B), many well-defined spectral lines are clearly observed. Three dominant spectral-peak frequency modes for each phase are shown in Table 1, with corresponding periods and intensities (powers) of the peaks. For each phase, an LSF curve (Eq (3)) to the residual data (Fig 1D) was calculated with three dominant periods listed in Table 1. The LSF curves thus obtained for phases I and II were indicated in Fig 3A and 3B, respectively. In each figure, the LSF curve reproduces the residual data well. Thus, the periods detected by MEM spectral analysis for each phase (Fig 2A and 2B, Table 1) were confirmed to be accurate. The good fit of each LSF curve to the residual data (Fig 3A and 3B) was supported by the fact that the ρ values between the residual data and the LSF curve were large. The values of ρ were 0.86 and 0.79 for phases I and II, respectively.

Close-ups of the low-frequency regions of the PSDs ($f \leq 6.0$) in Fig 2A and 2B are shown in Fig 4A and 4B, respectively. For phase I (Fig 4A), the dominant spectral peak is observed at $f = 2.9$ (0.34-year) with considerably large powers representing the amplitude of $x(t)$ at each frequency. For phase II (Fig 4B), the dominant spectral peak is observed at $f = 2.2$ (0.45-year) with considerably large powers. For phase I, the spectral peak at $f = 2.9$ corresponds to the period 4.1-month, which reflects intervals of four large waves of the COVID-19 pandemic with peaks during April 2020 –May 2021 observed in the inset of Fig 1A. For phase II, the spectral peak at $f = 2.2$ corresponds to the period 5.4-month, which reflects intervals of four large waves of the COVID-19 pandemic with peaks during August 2021 –January 2023 observed in Fig 1A.

## Segment time series analysis

A 3D spectral array obtained by segment time series analysis is shown in Fig 5, where power is plotted versus frequency (abscissa) and time (right-hand ordinate). In the frequency range of

**Table 1. Frequencies and powers of the three dominant spectral peaks shown in Fig 2A and 2b.** (a) Phase I (pre-Tokyo Olympic Games, 11 February 2020–22 July 2021). (b) Phase II (post- Tokyo Olympic Games, 23 July 2021–21 February 2023).

| Variable | $f$ (1/year) | Power |
|---|---|---|
| Phase I | 0.6 | 0.83 |
|  | 2.9 | 0.51 |
|  | 3.8 | 0.16 |
| Phase II | 0.6 | 0.87 |
|  | 1.6 | 0.80 |
|  | 2.2 | 1.13 |

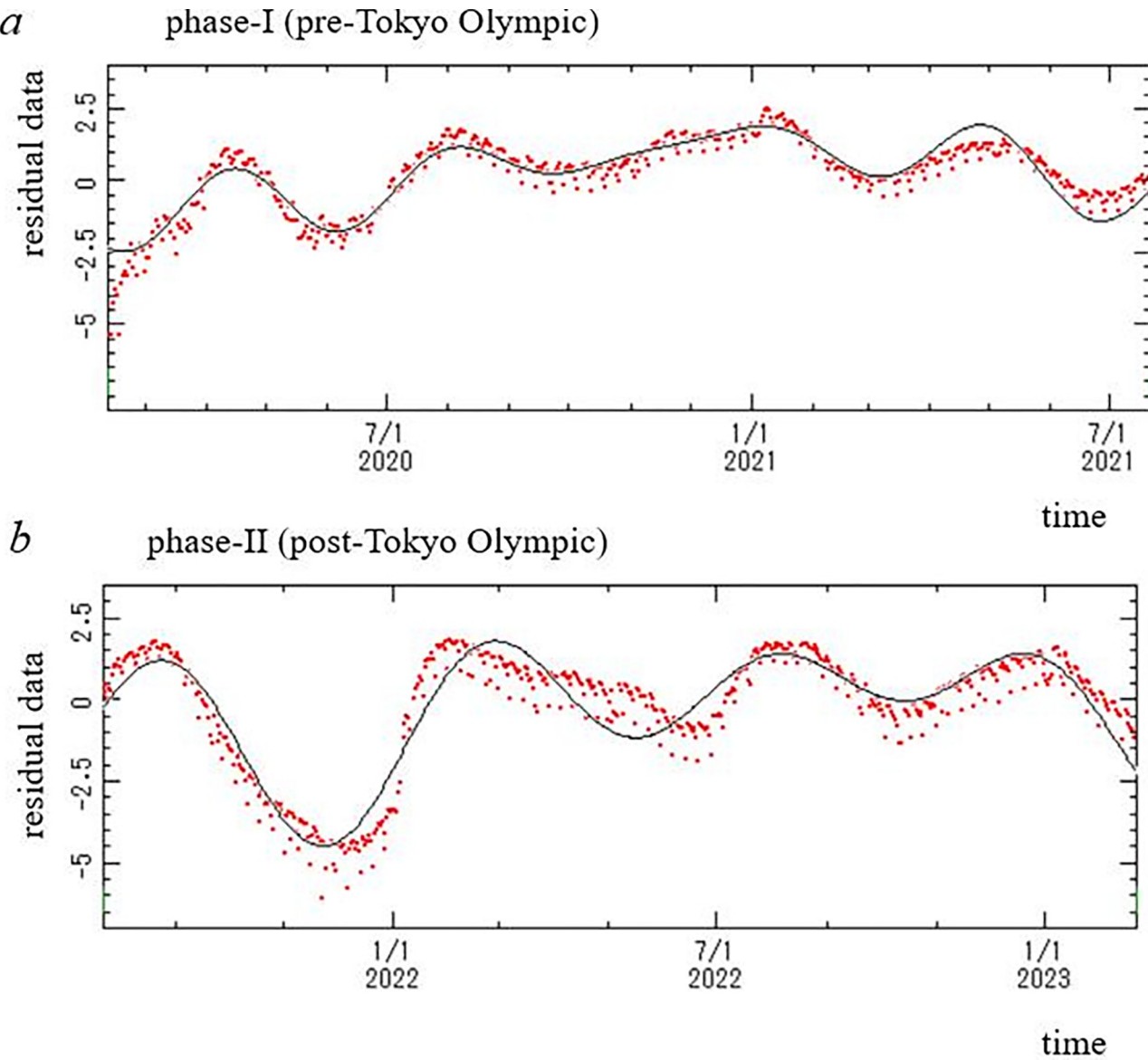

**Fig 3. Comparison of the least squares fitting curve (solid line) with the residual data (red dot).** (a) Phase I (pre-Tokyo Olympic Games, 11 February 2020–22 July 2021). (b) Phase II (post- Tokyo Olympic Games, 23 July 2021–21 February 2023).

$1.0 \leq f \leq 10.0$ (corresponding to 0.25-year to 1 year), the spectral lines are clearly observed over the entire time range.

The temporal variations of the frequency of dominant spectral lines observed in the 3D spectral array in the frequency range of $1.0 \leq f \leq 4.0$ (Fig 5) are plotted in Fig 6. As seen in the figure, spectral peaks were observed around $f = 3.0$ (0.33 year) before May 2021. After May 2021, and beginning before the Tokyo Olympic Games started in July 2021, spectral peaks gradually migrated to the low-frequency range, and were observed to be around $f = 2.2$ from July to November 2021, and then remained relatively constant around $f = 2.0$ thereafter.

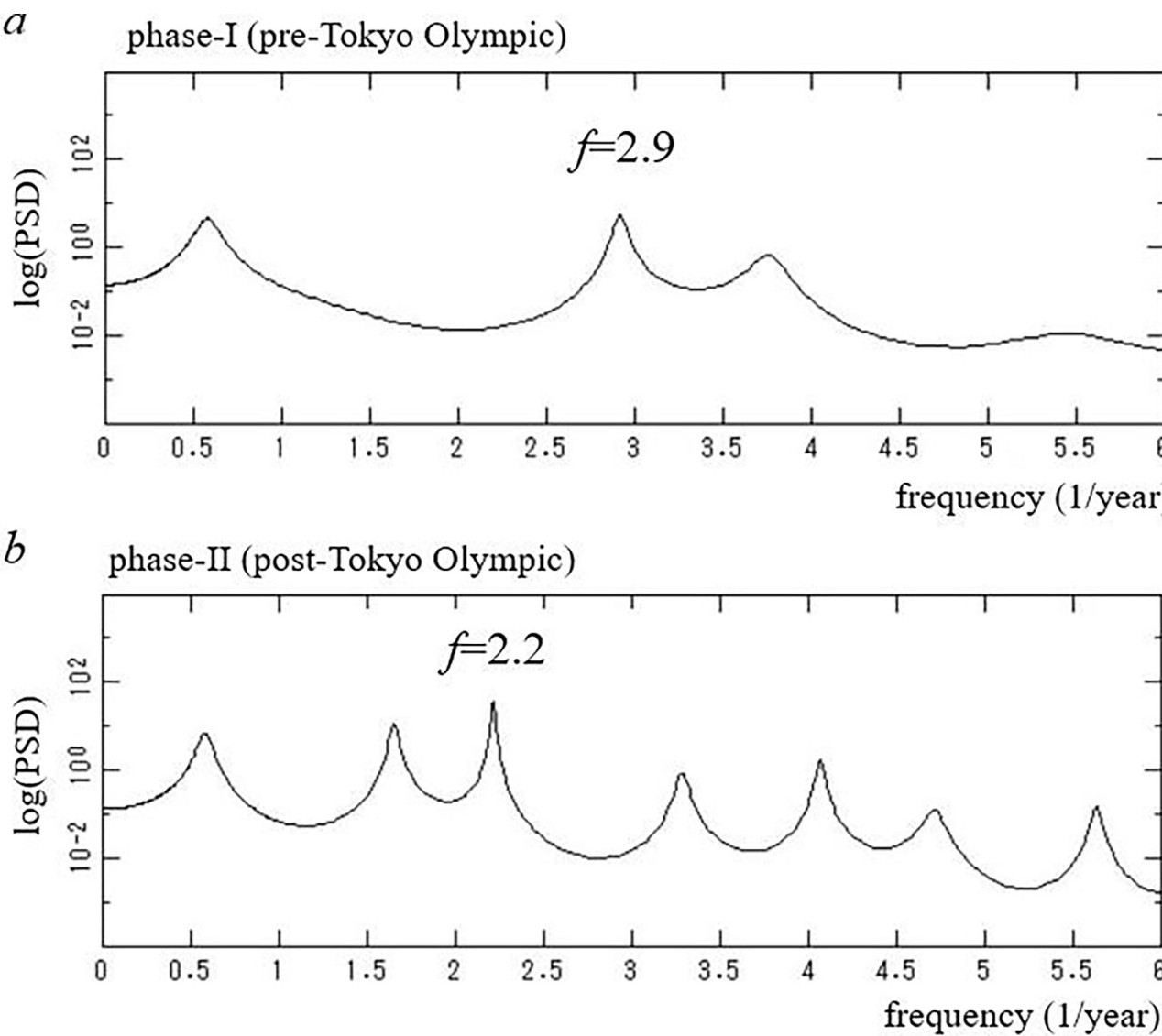

**Fig 4. Close-up of the low-frequency region ($f \leq 6.0$) in Fig 2A and 2B.** (a) Phase I (pre-Tokyo Olympic Games, 11 February 2020–22 July 2021). (b) Phase II (post- Tokyo Olympic Games, 23 July 2021–21 February 2023).

## Discussion

The most notable result obtained in the present study is that, as observed in Fig 6, the gradual migration of the spectral line to the low-frequency range from 3.0 (0.33-year) to 2.0 (0.5-year) during May to July 2021 is already observable before the commencement of the Tokyo Olympic Games in July 2021 and immediately after the introduction of mass and workplace vaccination programs in April 2021, at a time when Japan's vaccination rate was 4%. The vaccination coverage increased rapidly from May, with a maximum of 1.6 million doses per day. By October 2021, more than 77 million people, equivalent to 61.8% of the targeted population, had completed the vaccination series [18]. The temporal behavior of periodic structures observed in Fig 6 indicates that theoretical studies for measles control programs, based on predictions that vaccination generates an increase in the inter-epidemic period (IEP), corresponding to the interval between major waves of an epidemic, may also apply to COVID-19 [19,20]. The

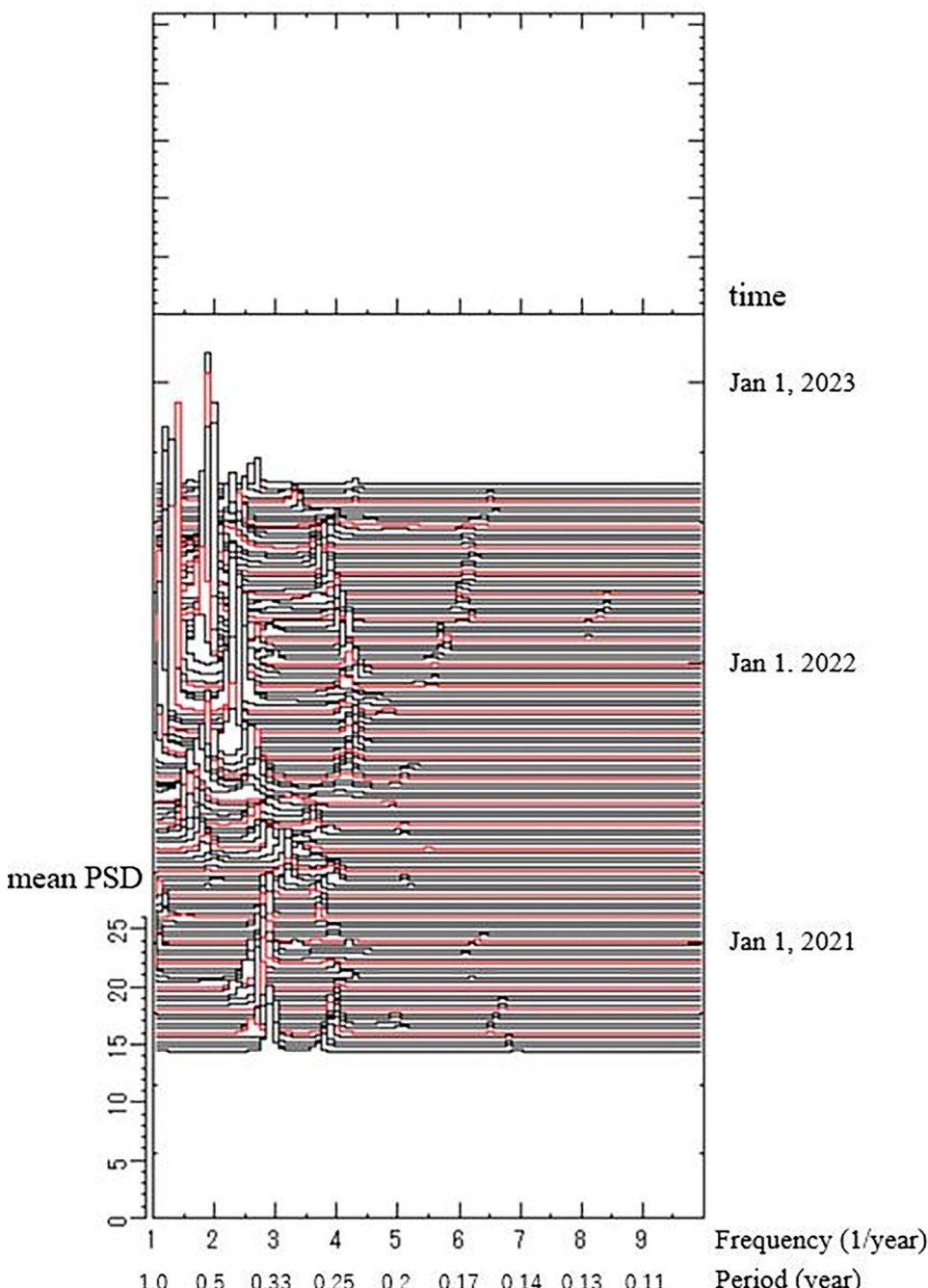

**Fig 5. Three-dimensional spectral array for the residual data in the frequency range of $1.0 \leq f \leq 10.0$.**

IEP of measles epidemics has been investigated with time series analysis and mathematical models [21–27]. The IEP represents the time required to accumulate a cohort of susceptible individuals large enough to effectively spread the measles virus throughout the community in the event of an external introduction of the virus. Our previous work investigated the IEP of measles epidemics in Japan and Wuhan in China using the present method of spectral analysis [19,20], and confirmed that the IEP increases as the vaccination ratio increases, as predicted by theoretical studies for a mathematical model of temporal variations of infectious diseases

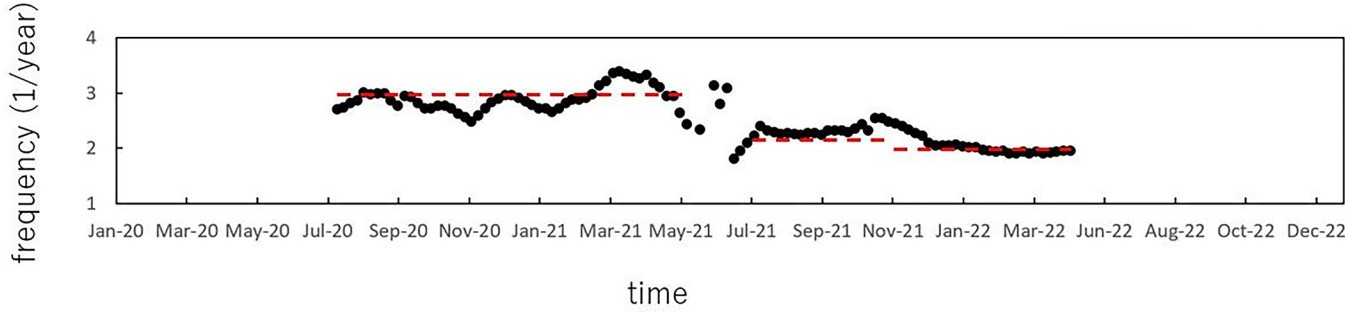

**Fig 6. Temporal variations of the frequencies of dominant spectral peaks detected in the frequency range of 1.0 $\leq f \leq$ 4.0.** The three red dashed lines indicate $f$ = 3.0 before May 2021, $f$ = 2.2 from July to November 2021, and $f$ = 2.0 thereafter.

[21,22,28]. Based on the theoretical studies of measles and our preceding work in that disease, the present finding that period structures of the COVID-19 data of Japan changed temporally after May 2021 (Fig 6) may be the effect of the increased vaccination rate in the previous month, April 2021.

With respect to the exponential characteristics of the PSDs for COVID-19 data (Fig 2A and 2B), our preceding work clarified that the PSDs for the time series generated from deterministic, nonlinear dynamical systems, such as the so-called susceptible/exposed/infectious/recovered (SEIR) epidemic model [29] and the Rössler, Lorenz and Duffing models [30,31], exhibit exponential characteristics. With respect to infectious disease epidemics, preceding research has confirmed that exponential spectral peaks are observed for incidence data of measles in Japan [15], Wuhan [19], New York City [29] and several communities in Denmark, the UK and the USA [32], as well as for chaotic and periodic time series generated by the SEIR epidemic model [29]. Thus, the present finding of exponential characteristics of the PSDs for COVID-19 (Fig 2A and 2B) suggests that the reported data for new positive cases of COVID-19 in Japan is dominated by deterministic nonlinear dynamics.

Regarding the magnitude of the PSD gradient λ, we found, based on previous studies on the SEIR model [29] and the Rössler model [30], that λ decreases from the periodic state through a bifurcation process and reaches a minimum in the chaotic state. The decrease in the magnitude of $\lambda$ can be considered to be the result of fluctuations mixed in a deterministic, nonlinear dynamical system [15]. Regarding the fluctuations, the author's group postulates two possibilities [15,30]. Firstly, the amplitude fluctuations could be caused by instabilities due to system non-linearity, as shown in the results using the Rössler model, and secondly, the fluctuations could be due to non-deterministic noise. In both cases, the magnitude of λ decreases because the high frequency component does not decrease as rapidly [15,33]. In the present study, we confirmed that the magnitude of λ for the pre-Olympics period is smaller than that of the post-Olympics period. This result reflects the relatively high complex variations of the data in the pre-Olympics period, which appears to support Sapkota et al.'s finding that, among Japan's 47 prefectures, the number of prefectures exhibiting chaotic characteristics was lower after the Tokyo Olympics than before [8]. A detailed study investigating chaotic characteristics for each prefecture in Japan is in the preparation phase.

To investigate the temporal variations of the COVID-19 pandemic, studies have been conducted using interrupted time series analysis [34,35] and the autoregressive integrated moving average (ARIMA) model [36]. Another important approach of time series analysis is the Bayesian spectral estimation method [37]. By contrast, in other studies, researchers interpreted the temporal variations of the COVID-19 pandemic using the SEIR model [38,39], which is a well-known nonlinear dynamical system for epidemics of infectious diseases [40]. However, the

interrupted time series analysis and ARIMA model using random noise, and the Bayesian spectral estimation method assuming approximate linearity of the time series data of COVID-19 cases, have a weakness in the interpretation of multiple periodicities of the time series data (Table 1) with characteristic fluctuations caused by nonlinear dynamics. The method Sapkota et al. applied to the time-series data of COVID-19 in Japan [8], the 0–1 test, is useful for examining whether the time series data is chaotic; however, this method is not suitable for detecting the multiple periodicities of the time series data (Table 1). In contrast, the present method based on MEM spectral analysis made it possible to identify periodicities in short time series with a high degree of frequency resolution [16]. In particular, the results obtained from segment time series analysis in the present study (Figs 5 and 6) showed that the periodic structures of the COVID-19 pandemic experience is changing over time.

The validity of the periodic modes detected by MEM spectral analysis (Table 1) was confirmed by calculating the LSF curve to $x(t)$ (Fig 3), which was expressed in the form of a linear combination of cosine functions in Eq (3). This curve is obtained from the Fourier transformation of the MEM-PSD, as described in Reference 16; that is, the following expression is obtained for the time series data:

$$x(t) = \sum_{j=1}^{N} \cos\{2\pi f_j(t + \theta_j)\} + \varepsilon(t), \tag{5}$$

where $f_j$ is the center frequency of the $j$-th peak intensity in MEM-PSD. The first term is identical to Eq (3), while the second term is undeterministic noise due to aperiodic or periodic fluctuations that cannot be expressed in the first term. This is the reason that the validity of the MEM spectral analysis results was confirmed by calculating the LSF curve with Eq (3) to $x(t)$ with MEM-estimated periods. In this study, the good fit of the LSF curves with Eq (3) to the residual data (Fig 3A and 3B) indicates that the periodic structure of the residual data (Table 1) was determined precisely. The conversion of analyzed time series data to normal distribution (Fig 1D') also contributes to precise determination of the MEM-estimated periods (Fig 3A and 3B, Table 1).

## Conclusions

In general, life phenomena are non-stationary and non-linear, with complex transitions from one state to another. Based on the results obtained in this study, it can be said that the periodic structure of infectious disease epidemics, including COVID-19, is changing with time. It is not appropriate to treat entire time series containing such states. Therefore, in order to elucidate the temporal evolution of non-linear phenomena, it is desirable to deal with segments of time series with short data lengths by segment time series analysis, as was done in this study. Investigation of the temporal variations of disease epidemics by segment time series analysis is expected to contribute to a long-term and effective COVID-19 control programs in Japan.

In conclusion, the following three results are confirmed in the present study: First, the exponential characteristics of PSDs can be observed for the COVID-19 data of Japan in both pre- and post-Olympics periods, which is characteristic of the nonlinear dynamical process. Second, the magnitude of the gradient of exponential PSD for the pre-Olympics period is smaller than that of the post-Olympics period, because of the relatively high complex variations of the data in the pre-Olympics period caused by a deterministic, nonlinear dynamical system and/or undeterministic noise. Third, changes in the periodic structures of the COVID-19 data were already occurring before the Tokyo Olympic Games began in July 2021 and immediately after the mass and workplace vaccination programs were introduced in April

2021, indicating that the findings of theoretical studies for measles control programs may also apply to the COVID-19 data.

## Supporting information

**S1 Dataset. Time series data of the daily reported number of new positive cases of COVID-19 for the entire country of Japan from 16 January 2020 to 21 February 2023.**
(ZIP)

## Acknowledgments

We thank Edanz (https://jp.edanz.com/ac) for editing a draft of this manuscript.

## Author Contributions

**Conceptualization:** Ayako Sumi.

**Data curation:** Ayako Sumi.

**Formal analysis:** Ayako Sumi.

**Funding acquisition:** Ayako Sumi.

**Investigation:** Ayako Sumi.

**Methodology:** Ayako Sumi.

**Project administration:** Ayako Sumi.

**Resources:** Ayako Sumi.

**Software:** Ayako Sumi.

**Supervision:** Ayako Sumi.

**Validation:** Ayako Sumi.

**Visualization:** Ayako Sumi.

**Writing – original draft:** Ayako Sumi.

**Writing – review & editing:** Ayako Sumi.

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
