## [Decision Letter · Decision Letter 0]

26 Jun 2023

PONE-D-23-11549Time series analysis of daily data of COVID-19 reported cases in Japan from January 2020 to February 2023PLOS ONE

Dear Dr. Sumi,

Thank you for submitting your manuscript to PLOS ONE. After careful consideration, we feel that it has merit but does not fully meet PLOS ONE’s publication criteria as it currently stands. Therefore, we invite you to submit a revised version of the manuscript that addresses the points raised during the review process. Please submit your revised manuscript by 15th July. If you will need more time than this to complete your revisions, please reply to this message or contact the journal office at plosone@plos.org. Please include the following items when submitting your revised manuscript:A rebuttal letter that responds to each point raised by the academic editor and reviewer(s). You should upload this letter as a separate file labeled 'Response to Reviewers'.A marked-up copy of your manuscript that highlights changes made to the original version. You should upload this as a separate file labeled 'Revised Manuscript with Track Changes'.An unmarked version of your revised paper without tracked changes. You should upload this as a separate file labeled 'Manuscript'.

We look forward to receiving your revised manuscript.

Kind regards,

Junyuan Yang

Academic Editor

PLOS ONE

Journal Requirements:

https://pubmed.ncbi.nlm.nih.gov/34893009/

https://www.cambridge.org/core/journals/epidemiology-and-infection/article/study-on-the-effect-of-measles-control-programmes-on-periodic-structures-of-disease-epidemics-in-a-large-chinese-city/82F89CE78E1DB999407272A6F29856FD

https://iopscience.iop.org/article/10.1143/JJAP.42.721

In your revision ensure you cite all your sources (including your own works), and quote or rephrase any duplicated text outside the methods section. Further consideration is dependent on these concerns being addressed.

“no”

Reviewers' comments:

Reviewer's Responses to Questions

**Comments to the Author**

1. Is the manuscript technically sound, and do the data support the conclusions?

Reviewer #1: Yes

Reviewer #2: Yes

Reviewer #3: Yes

2. Has the statistical analysis been performed appropriately and rigorously? 

Reviewer #1: Yes

Reviewer #2: Yes

Reviewer #3: Yes

3. Have the authors made all data underlying the findings in their manuscript fully available?

Reviewer #1: Yes

Reviewer #2: Yes

Reviewer #3: Yes

4. Is the manuscript presented in an intelligible fashion and written in standard English?

Reviewer #1: Yes

Reviewer #2: Yes

Reviewer #3: Yes

5. Review Comments to the Author

Reviewer #1: Dear authors,

It has been a pleasure to review the manuscript "Time series analysis of daily data of COVID-19 reported cases in Japan from January 2020 to February 2023".

The paper is technically sound, and presents a time series analysis, using maximum entropy methods for spectrum estimation. Using this method, the authors look at changes on the dynamics of COVID-19 before and after the Tokyo Olympics

Please find below some comments and suggestions. The comments are to clarify certain points of the manuscript; the minor comments aim to improve the readibility.

Comments:

- Why is the MEM-PSD method not limited by the sampling frequency in the data? A brief explanation would help the reader.

- Fig 1, and page 9 line 145: The histogram in 1e does not look Gaussian, although it certainly looks better than the original data. Please discuss what limitations could the study have (or how the results could be impacted) by the non-Gaussianity.

- "Results" section: MEM-PSD is applied several times, either to data or to residuals. A more detailed description of the approach, in order that is applied should be provided, otherwise the reader is confused about where MEM is applied.

- The preparation of the data could be moved to the methods part to make the results section clearer.

- Page 10, line 165: how is lambda estimated?

- Page 11, line 182: the method is more commonly now as "segmented time series analysis". The method should be presented in the methods section.

- Page 14, line 239: rather than "is based", one would say "is dominated" or similar; real data is certainly not based on a model!

- Page 15, line 255: Using "study" twice is confusing. Change to "A detailed study investigating chaotic characteristics for each prefecture in Japan is in preparation", or similar.

- Page 16, line 270: characteristic (rather than peculiar).

Minor comments:

Page 2, line 19 (and elsewhere): variational typically refers to optimization for functionals; here a more common expression would be "variation" or "variability", or even just "temporal estructures".

Page 2, line 27: The sentence about measles control needs a little bit of context.

Page 3, line 34: "recurrently" (not recursively)

Page 4, line 61: Could you mention the total population (or what percentage of the total population the total number of cases represents)?

Reviewer #2: The current submission performs the time series analysis on the COVID-19 pandemic date in Japan. The analysis confirms that the power spectral densities for data in both the pre- and post-Tokyo Olympics periods show exponential characteristics.

Following are some points for further consideration:

(a) The main concern: There have been extensive studies on time series analysis for daily COVID-19 data for almost every region with different methodologies. The manuscript fails to present more details to compare the methodology in the current with other approaches, as well as the results of the current study and other existing ones for Japanese pandemic pattern.

(b) The data in the supplementary material are daily incidence, however, it was mentioned in L82 that "monthly time series data". Overall, different time units are used, including day (daily date), month and year (for frequency and period).

(c) formula in Eq ([Disp-formula pone.0285237.e004]) of L165 is not appropriately shown

Reviewer #3: A meticulously written paper offering a fresh perspective on the analysis of short time series to detect temporal variations is presented.

The following comments have been noted:

Concerning line 118, it appears that the date mentioned, June 30, 2023, may be a typographical error and should likely be corrected to 2021.

It is suggested to consider presenting Figure 1b as an inset within Figure 1a, if feasible.

In Figure 1e, it is recommended to include a vertical dashed line to delineate Phase I and Phase II. Additionally, annotating the phases would enhance clarity and understanding.

Regarding Figure 2a and 2b, annotating Phase I and Phase II directly within the figures would be helpful. Furthermore, the presence of two lines for low frequencies in both figures raises questions about their representation. It is advised to address this in the text by explaining whether they indicate a specific trend or any other relevant information.

For Figure 3a and 3b, where f = 2.8 and 2.0, it is suggested to provide explanations in the text regarding these values in the context of the original time series of daily COVID data. Specifically, clarifying whether the time series data represents new cases or includes repeat cases would be valuable information that seems to be missing from the text.

Enhancing the labels and axes in Figure 4 is recommended to improve readability and comprehensibility.

Notably, Figure 5 is considered the crux of the paper. To enhance its presentation, the addition of three horizontal dashed lines representing different means is proposed. Consistency can be maintained by using the blue color scheme used throughout the paper.

Regarding the statement made in lines 95-97 about confirming the validity of MEM spectral analysis results through least squares fitting (LSF) curve calculation pertaining to the original time series data x(t) with MEM-estimated periods, it would be beneficial to support this claim by referencing other works. Alternatively, if other comparative methods or known signals are not available for this specific dataset, it would be appropriate to suggest alternative methods like comparing with results from other analysis techniques such as Fourier analysis (e.g., periodograms, Welch's method).

The discussion section is highly commendable, and it is encouraged to establish connections between the findings and real-world data to underscore the practical implications of the study, especially in the field of epidemiology.

6. PLOS authors have the option to publish the peer review history of their article (what does this mean?). If published, this will include your full peer review and any attached files.

Reviewer #1: No

Reviewer #2: No

Reviewer #3: **Yes: **Nabin Sapkota

---

## [Author Response · Author response to Decision Letter 0]

8 Aug 2023

Reviewer 1: I have incorporated all of your suggestions in my revision. They were very helpful. Thank you. 

Reviewer 2: I have incorporated all of your suggestions in my revision. They were very helpful. Thank you. 

Reviewer 3: I have incorporated all of your suggestions in my revision. They were very helpful. Thank you.

---

## [Decision Letter · Decision Letter 1]

31 Aug 2023

Time series analysis of daily reported number of new positive cases of COVID-19 in Japan from January 2020 to February 2023

PONE-D-23-11549R1

Dear Dr. Sumi,

We’re pleased to inform you that your manuscript has been judged scientifically suitable for publication and will be formally accepted for publication once it meets all outstanding technical requirements.

Kind regards,

Junyuan Yang

Academic Editor

PLOS ONE

Additional Editor Comments (optional):

Reviewers' comments:

Reviewer's Responses to Questions

**Comments to the Author**

1. If the authors have adequately addressed your comments raised in a previous round of review and you feel that this manuscript is now acceptable for publication, you may indicate that here to bypass the “Comments to the Author” section, enter your conflict of interest statement in the “Confidential to Editor” section, and submit your "Accept" recommendation.

Reviewer #2: All comments have been addressed

Reviewer #3: All comments have been addressed

2. Is the manuscript technically sound, and do the data support the conclusions?

Reviewer #2: Yes

Reviewer #3: Yes

3. Has the statistical analysis been performed appropriately and rigorously? 

Reviewer #2: Yes

Reviewer #3: Yes

4. Have the authors made all data underlying the findings in their manuscript fully available?

Reviewer #2: Yes

Reviewer #3: Yes

5. Is the manuscript presented in an intelligible fashion and written in standard English?

Reviewer #2: Yes

Reviewer #3: Yes

6. Review Comments to the Author

Reviewer #2: The author has addressed all concerns raised. In particular, more comparisons on time series analysis have been included in the revised version. The referee has no further comments.

Reviewer #3: All my comments/suggestions were addressed. I hope this work may give ideas to other researchers in various others related or non-related fileds as well.

7. PLOS authors have the option to publish the peer review history of their article (what does this mean?). If published, this will include your full peer review and any attached files.

Reviewer #2: No

Reviewer #3: **Yes: **Nabin Sapkota

---

## [Editor Report · Acceptance letter]

7 Sep 2023

PONE-D-23-11549R1 

Time series analysis of daily reported number of new positive cases of COVID-19 in Japan from January 2020 to February 2023 

Dear Dr. Sumi:

I'm pleased to inform you that your manuscript has been deemed suitable for publication in PLOS ONE. Congratulations! Your manuscript is now with our production department. 

Kind regards, 

on behalf of

Dr. Junyuan Yang 

Academic Editor

PLOS ONE